# Hyperuricemia, a Non-Independent Component of Metabolic Syndrome, Only Predicts Renal Outcome in Chronic Kidney Disease Patients without Metabolic Syndrome or Diabetes

**DOI:** 10.3390/biomedicines10071719

**Published:** 2022-07-16

**Authors:** Sheng-Wen Niu, Hugo You-Hsien Lin, I-Ching Kuo, Yen-Yi Zhen, Eddy-Essen Chang, Feng-Ching Shen, Yi-Wen Chiu, Jer-Ming Chang, Chi-Chih Hung, Shang-Jyh Hwang

**Affiliations:** 1Graduate Institute of Clinical Medicine, College of Medicine, Kaohsiung Medical University, Kaohsiung 80708, Taiwan; 950138kmuh@gmail.com (S.-W.N.); 980135kmuh@gmail.com (I.-C.K.); 2Division of Nephrology, Department of Internal Medicine, Kaohsiung Medical University Hospital, Kaohsiung Medical University, Kaohsiung 80708, Taiwan; yukenlin@yahoo.com.tw (H.Y.-H.L.); a0928306234@gmail.com (Y.-Y.Z.); eotaxin002@yahoo.com.tw (E.-E.C.); sam0927035368@gmail.com (F.-C.S.); chiuyiwen@gmail.com (Y.-W.C.); jemich@kmu.edu.tw (J.-M.C.); 3Division of Nephrology, Department of Internal Medicine, Kaohsiung Municipal Ta-Tung Hospital, Kaohsiung Medical University, Kaohsiung 80145, Taiwan; 4Department of Medicine, College of Medicine, Kaohsiung Medical University, Kaohsiung 80708, Taiwan; 5Department of Medical Research, Kaohsiung Medical University Hospital, Kaohsiung Medical University, Kaohsiung 80708, Taiwan; 6Regenerative Medicine and Cell Therapy Research Center, Kaohsiung Medical University, Kaohsiung 80708, Taiwan

**Keywords:** hyperuricemia, renal outcome, metabolic syndrome, diabetes, chronic kidney disease

## Abstract

Uric acid (UA) is elevated in metabolic syndrome (MS) and diabetes (DM). UA is associated with central obesity and blood glucose and is proposed as a criterion of MS. Previous reports showed that UA could predict renal outcome in CKD. However, recent clinical trials did not demonstrate the benefits of urate-lowering agents (ULA) for renal outcome. Whether the prognostic value of UA for renal outcome is independent of MS or secondary to MS in CKD patients is unknown. Our study included 2500 CKD stage 1–4 Asian patients divided by UA tertiles and MS/DM. In linear regression, UA was associated with obesity, C-reactive protein, and renal function. In Cox regression, high UA was associated with worse renal outcome in non-MS/DM, but not in MS/DM: hazard ratio (95% confidence interval) of UA tertile 3 was 3.86 (1.87–7.97) in non-MS/DM and 1.00 (0.77–1.30) in MS/DM (*p* for interaction < 0.05). MS was associated with worse renal outcome, but redefined MS (including hyperuricemia as the 6th criteria) was not. In conclusion, hyperuricemia is associated with worse renal outcome in non-MS/DM and is not an independent component of MS in CKD stage 1–4 patients. Hyperuricemia secondary to MS could not predict renal outcome.

## 1. Introduction

Uric acid (UA) is elevated in metabolic syndrome (MS) and diabetes (DM). The prevalence of hyperuricemia in MS and DM is approximately 7.7 times higher than that in the general population [1]. On the other hand, the prevalence of MS and DM is also high in hyperuricemic patients [2].

Hyperuricemia is accompanied by hyperglycemia, which is attributed to increased reabsorption of glucose as well as UA from GLUT9 by elevated insulin resistance (IR) [3,4]. Our previous report demonstrated that amelioration of renal reabsorption of UA via benzbromarone (a uricosuric agent) is associated with a lower incidence of diabetes [5]. J. Sun et al. unveiled a significant positive association between total percent fat and serum UA, especially in the male population [6]. V. Karava et al. found that even children with high relative fat mass including normal weight obesity are at risk for IR regardless of chronic kidney disease (CKD) stage [7]. We also noted that hyperuricemia is accompanied by not only central obesity [8] but hypertriglyceridemia [9] and is proposed as a criterion of MS. UA could inhibit adenosine monophosphate kinase (AMPK) [10] and tilt the balance toward metabolic syndrome [11]. Metformin (an AMPK activator) can lower glucose and UA. Peroxisome proliferator-activated receptors (PPARs) are also involved in metabolic syndrome. Our previous reports showed that pioglitazone (a PPARγ agonist) could lower glucose, lipids, and the incidence of gout [12].

UA is associated with the progression of kidney disease in basic and clinical studies [13]. UA increases renal inflammation and fibrosis [14], and UA crystals cause gouty nephropathy in basic studies [15,16]. Some observational studies have demonstrated the detrimental effect of hyperuricemia in CKD patients [17], but some studies disagree [18]. One explanation is that the progression of CKD elevates UA via reduced renal excretion [19].

UA-lowering therapy did not improve renal outcome in recent large randomized controlled trials and meta-analyses in CKD patients with hyperuricemia [20,21,22,23]. The interrelationship between UA and MS/DM discussed above suggests that hyperuricemia could be secondary to MS components [24]. Thus, we hypothesize that hyperuricemia is an independent factor for worse renal outcome only in non-MS/DM and is not an independent component of MS in the CKD population. We tested whether MS/DM modifies the association between UA and renal outcome in CKD stage 1–4 patients.

## 2. Materials and Methods

### 2.1. Study Design and Participants

As described previously [25], two affiliated hospitals of Kaohsiung Medical University in southern Taiwan, participated in the Kaohsiung CKD Delayed Dialysis Comprehensive Care Program Study from 11 November 2002 to 31 May 2009, and followed up until 31 December 2014. Patients with CKD stages 1–5 who did not receive renal replacement therapy were included as inclusion criteria (N = 3659) at first, and (1) acute kidney injury, defined as >50% decrease in estimated glomerular filtration rate (eGFR) within 3 months (N = 62), (2) lost to follow-up in less than 3 months (N = 90), (3) chronic kidney disease stage 5 (N = 1007), was excluded, then finally this analysis included 2500 patients with CKD stages 1–4 (Figure 1). All patients will be followed up for at least 3000 days if there is no death. Informed consent was given to all participating patients. The study protocol was approved by the Institutional Review Board of Kaohsiung Medical University Hospital with approval number KMUH-IRB-990198.

### 2.2. Collection of Demographic, Medical, and Laboratory Data

Baseline variables included demographic characteristics (age and gender), laboratory data (eGFR, hemoglobin [Hb], albumin, C-reactive protein [CRP], total cholesterol, and triglycerides), medical history (diabetes, hypertension, cardiovascular disease (CVD, including coronary artery disease and cerebrovascular disease), Charlson comorbidity index score, metabolic syndrome and malnutrition-inflammation score [MIS]). Demographic characteristics formed baseline records, and medical histories were obtained through physician review of medical records and interviews with patients. BMI is calculated by dividing weight (in kilograms) by the square of height (in meters). Hip and waist circumferences were measured using WHO protocols. Mean arterial pressure was calculated from mean systolic and diastolic blood pressures measured 3 months before and after enrollment using the formula of one-third of mean systolic blood pressure plus two-thirds of mean diastolic blood pressure. Urine protein to creatinine ratio (Upcr) was obtained by dividing urine protein (mg) by urine creatinine (g) in random spot urine samples. According to the protocol, biochemical measurements were taken during the screening visit, at the baseline visit, and then every 3 months. 

### 2.3. Diagnosis of Metabolic Syndrome and Diabetes

Diagnosing MS according to the American Heart Association/National Heart, Lung and Blood Institute criteria. When a person has any of the following five abnormalities greater than or equal to 3: (Waist circumference greater than or equal to 102 cm for men, and greater than or equal to 88 cm for women, Asian American waist circumference greater than or equal to 90 cm for men and greater than or equal to 80 cm for women). Hypertriglyceridemia (triglycerides greater than or equal to 150 mg/dL or subject is taking lipid-controlling medications), low HDL (HDL; less than 40 mg/dL in men, less than 50 mg/dL in women, or subject is taking lipid-controlling medications), hypertension (systolic blood pressure greater than or equal to 130 mmHg/diastolic blood pressure greater than or equal to 85 mmHg, or subject is taking hypertension medication), and hyperglycemia (fasting blood glucose greater than or equal to 100 mg/dL or subject is taking diabetes medication) [26]. Since Taiwan is an Asian country, central obesity in Taiwanese national MS criteria is defined by the Taiwan Department of Health as a waist circumference greater than or equal to 90 cm for men and a waist circumference greater than or equal to 80 cm for women [27]. In addition, HDL is not included in the parameters of health examinations mandated by occupational health regulations; therefore, high total cholesterol (greater than or equal to 200 mg/dL or taking a lipid-lowering drug agent) is the criterion in lieu of a low HDL value.

The diagnosis of diabetes was previously defined according to the 1999 World Health Organization (WHO) diagnostic criteria, fasting blood glucose ≥ 126 mg/dL, and two-hour blood glucose (postprandial) ≥ 200 mg/dL [28]. According to the American Diabetes Association (ADA) guidelines, an HbA1C level of 6.5% or higher is also in the range of diabetes [29].

### 2.4. Outcomes

Hemodialysis, peritoneal dialysis, or transplantation was defined as renal replacement therapy. Reviewing death certificates using charts or the National Death Index was ascertained for all-cause mortality. Record all-cause mortality before and after renal replacement therapy.

### 2.5. Statistical Analysis

All patients were stratified by MS/DM and uric acid tertiles and expressed as a percentage of categorical data. The mean ± standard deviation in a continuous variable has an approximately normal distribution. The median and interquartile range of continuous variables have skewed distributions. Chi-square test was carried out to test for differences between groups when encountering categorical variables. One-way analysis of variance was carried out for different groups of continuous variables. Cox proportional hazards analysis was used to study the relationship between uric acid and clinical outcomes. Skewed distributed continuous variables are log-transformed to obtain a normal distribution. Consistent with our previous paper, covariates were selected based on clinical relevance [30]. Adjusted covariates included age, sex, eGFR, log (UPCR), DM, CVD, cancer, liver disease, HbA1c, smoking status, hemoglobin, albumin, log (CRP), log (cholesterol), phosphorus, and BMI. Statistical analysis was performed using two legally licensed software, STATA 15.0 and SPSS version 20.0 for Windows (SPSS Inc., Chicago, IL, USA).

## 3. Results

### 3.1. Patient Characteristics by MS/DM and UA Tertiles

Comparing 1872 MS/DM patients with 628 non-MS/DM patients, the MS/DM group had older age, a higher percentage of females, a higher percentage of CVD, lower eGFR, higher proteinuria, and worse clinical outcomes (Table 1). The MS/DM group also had a higher percentage of all MS components (Table 2), indicating a significant difference between the MS/DM and non-MS/DM groups, as shown in Table 1 and Table 2. 

In both the non-MS/DM and MS/DM groups, UA tertiles were positively associated with age, diuretic use, waist circumference, diastolic blood pressure, triglycerides, and phosphorus and negatively associated with eGFR and hemoglobin (Table 1 and Table 2). In both groups, UA tertile 3 had worse renal outcomes. In the non-MS/DM group, UA tertile 3, compared with UA tertile 1, had a lower percentage of females, higher systolic blood pressure, and a higher percentage of hypertension, but these trends were not observed in the MS/DM group. In contrast, in the MS/DM group, compared with UA tertile 1, UA tertile 3 had a higher percentage of CVD, higher proteinuria, a higher percentage of 3 MS components (waist, HDL cholesterol, and triglycerides), and worse composite outcomes of RRT and mortality, but these trends were not observed in the non-MS/DM group. 

To study whether UA could be a component of criteria in MS, we compared the similarity and difference between UA tertile 3 in the non-MS/DM group and all MS/DM groups as indicated by $ in Table 1 and Table 2. Although both groups had similar percentages of RRT, UA tertile 3 in the non-MS/DM group had better data for all 5 MS components.

### 3.2. Multivariate Linear Regression for UA

The results of multivariate linear regression for UA (Table 3) indicated that males, lower eGFR, higher body mass index (BMI), higher CRP, higher phosphorus, longer waist circumference, higher triglycerides, and the use of ULA and diuretics had significantly higher uric acid levels. Although some parameters showed different trends of association with UA tertiles in Table 1 and Table 2, we did not find that MS/DM modified the association between UA and the parameters in Table 3.

### 3.3. UA Tertiles and Clinical Outcomes

In the non-MS/DM group, in the fully adjusted Cox regression (Table 4), a 1 mg/dL increase in UA was significantly related to a 27% increase in the risk of RRT (HR: 1.27, 95% CI: 1.10–1.47), a 13% increase in the risk of RRT + 50% eGFR decline (HR: 1.13, 95% CI: 1.04–1.24), and an 18% increase in the risk of RRT + mortality before RRT (HR: 1.18, 95% CI: 1.06–1.31) but was not related to all-cause mortality. UA tertile 3 had a higher risk of RRT (HR: 3.86, 95% CI: 1.87–7.97), RRT + 50% eGFR decline (HR: 1.69, 95% CI: 1.08-2.65), and RRT + mortality before RRT (HR: 1.73, 95% CI: 1.07–2.80) compared with UA tertile 1. 

In the MS/DM group, in the fully adjusted Cox regression (Table 4), a 1 mg/dL increase in UA was significantly related to a 5% increase in all-cause mortality (HR: 1.05, 95% CI: 1.00–1.10) but was not related to RRT, RRT + 50% eGFR decline, or RRT + mortality before RRT. UA tertile 3 had a marginally higher risk of RRT + mortality before RRT (HR: 1.20, 95% CI: 0.99–1.46, *p* = 0.060) compared with UA tertile 1.

### 3.4. MS and Clinical Outcomes

To study whether UA could be a component of MS, we compared the association between clinical outcomes and MS (defined by traditional 3 of 5 components or by UA as 6th components [3 of 6 components]) (Table 5). In the fully adjusted Cox regression, MS by traditional definition was related to a 73% increase in the risk of RRT (HR: 1.73, 95% CI: 1.24–2.43), a 35% increase in the risk of RRT + 50% eGFR decline (HR: 1.35, 95% CI: 1.12–1.63), and a 28% increase in the risk of RRT + mortality before RRT (HR: 1.28, 95% CI: 1.00–1.64) but was unrelated to all-cause mortality. By adding UA in the definition, MS was only marginally related to RRT (HR: 1.27, 95% CI: 0.94–1.71, *p* = 0.117) and did not relate to the other 3 outcomes.

## 4. Discussion

In CKD stage 1–4 patients, we demonstrated that uric acid was associated with diabetes and some components of MS, and the association between uric acid and RRT was modified by MS/DM. Hyperuricemia was associated with RRT or composite outcome in non-MS/DM but not in MS/DM. We further demonstrated that adding hyperuricemia as a 6th component of MS did not increase the prognostic value of MS. Both types of evidence suggest that hyperuricemia secondary to MS could not predict renal outcome. Therefore, future clinical trials of ULA for renal outcome should be conducted in non-MS/DM populations.

Hyperuricemia and gout are associated with the progression of CKD through crystal-dependent and crystal-independent mechanisms [31]. Monosodium urate crystals, which attract macrophages and stimulate vascular smooth muscle cell (VSMC) proliferation [32], can activate inflammasomes [33]. Proinflammatory cytokines (IL-1β and IL-18) [34] and active inflammasomes result in the development and progression of tubulointerstitial disease [35]. Hyperuricemia also induces renal vasoconstriction via inflammation, endothelial dysfunction, and renin–angiotensin system activation similar to its effect on cardiovascular outcomes [36].

Early observational studies demonstrated the detrimental effect of hyperuricemia in CKD [17]. Decreasing UA by uricosuric agents or xanthine oxidase inhibitors in hyperuricemic patients with CKD was proposed to slow renal function decline [37]. A retrospective epidemiologic cohort study of 12,751 patients demonstrated that UA-lowering therapy (allopurinol 97%; febuxostat 2%; probenecid 1%) in CKD was associated with a higher odds ratio for a 30% improvement in eGFR in CKD stage 2–3 but not in stage 4 [38]. A propensity score-matched cohort of CKD stage 3–4 patients with asymptomatic hyperuricemia in stage 3–4 CKD pointed out that UA-lowering therapy did not delay CKD progression [18]. A propensity score-matched study in incident gout patients demonstrated that febuxostat and benzbromarone both have better CKD recovery than allopurinol [39]. 

Recent large randomized controlled trials did not find a favorable effect of UA-lowering therapy. The Preventing Early Renal Loss in Diabetes (PERL) trial failed to confirm the clinically meaningful benefits of allopurinol on kidney outcomes among patients with type 1 DM and early-to-moderate diabetic kidney disease [20]. In the Febuxostat Versus Placebo Randomized Controlled Trial Regarding Reduced Renal Function in Patients With Hyperuricemia Complicated by Chronic Kidney Disease Stage 3 (FEATHER) study, febuxostat had no significant evidence to alleviate the decline in the eGFR in stage 3 CKD patients with hyperuricemia in the entire cohort [21]. Another RCT in Australia and New Zealand also revealed that in CKD patients with a high risk of progression, allopurinol did not lower the decline in eGFR compared with placebo [22]. One recent meta-analysis including 16 RCTs concluded insufficient evidence to support the renoprotective effects of the three urate-lowering agents in CKD patients with hyperuricemia [23]. Thus, we suggest non-MS/DM as a potential target group for UA-lowering agent therapy. 

Hyperuricemia is associated with every component of metabolic syndrome. A previous study found that the components of MS were elevated across the UA quartiles [40]. MS component 1 blood sugar: insulin resistance could increase both glucose and UA [3]; MS component 2 waist: central obesity could increase both waist circumference and UA [41]; MS component 3 triglyceride: abnormal fatty acid metabolism could increase both triglycerides and UA [42]; MS component 4 cholesterol: abnormal cholesterol metabolism could decrease HDL cholesterol and increase UA [43]. MS component 5 hypertension: endothelial dysfunction could increase both blood pressure and UA [44]. All of the above comorbidities could lead to CKD progression and finally mortality [45]. In our data, MS by the traditional definition was associated with RRT, while MS by adding UA in the definition was only marginally related to RRT. These results suggested that MS components, instead of UA, could influence renal outcome. 

MS component 1 blood sugar: hyperuricemia could be secondary to insulin resistance as in metabolic syndrome and diabetes. Mechanistically, elevated insulin resistance ameliorates renal UA excretion by stimulating UA-anion exchanger UA transporter 1 (URAT1, as SLC22A12) [3]. Studies have pointed out a significant correlation between uric acid levels and homeostatic model assessment (HOMA) scores in patients with obesity or at risk for T2DM [46,47]. One study of 6027 nondiabetic individuals demonstrated that serum UA is independently associated with impaired fasting glucose (IFG) and insulin resistance but not with beta-cell dysfunction [48]. A study of 174,695 adults without self-reported use of ULA, hypoglycemic agents, or lipid-lowering drugs demonstrated that noninsulin-based IR indices were significantly associated with hyperuricemia [49]. 

MS component 2 waist: hyperuricemia could be secondary to obesity. Obesity is positively associated with hyperuricemia by inducing UA production and reducing renal UA excretion [41]. Obese adipose tissue is hypoxic, which enhances dysfunction of adipocytokines, followed by chronic inflammation [50] and increases UA production [51]. PPARγ inhibitor (pioglitazone) treatment is associated with a decreased incidence of gout [12]. Obese adipose tissue is symbolized by the enhancement of fatty acid (FA) synthesis, and FA synthesis is presumably closely associated with de novo purine synthesis through the induction of the XOR-related pathway [52], followed by hyperuricemia, glucose dysmetabolism, and a prothrombotic state [53]. Our results showed the association of BMI and waist circumference with uric acid and are in line with past assumptions and studies. 

MS components 3 and 4 triglyceride and cholesterol: hyperuricemia could be secondary to dyslipidemia [43]. The lipid profiles are closely related to UA [54]. A previous study indicated that lowering hypertriglyceridemia with statins could improve hyperuricemia at the same time [55], followed by a reduced incidence of MACEs [56]. Our results showed that the association of triglyceride with uric acid is consistent with these studies. 

MS component 5 hypertension: The association between hypertension and hyperuricemia has been noted for a long time; 25–47% of untreated hypertensive adults had hyperuricemia, and 75% had malignant hypertension in an early study [57]. In MS patients, hyperuricemia was also considered an independent factor associated with the development of elevated BP in young adults [58]. Hyperuricemia is also an independent risk factor for cardiovascular disease [59]. In the present study, CRP was associated with higher UA, while hypertension and blood pressure were not. Pathways of immune system activation and inflammation implicated in UA are involved in the development of vascular inflammation [60], further promoting endothelial dysfunction and VSMC proliferation [44]. These data suggested that hypertension is not secondary to UA in this population and that vascular inflammation might be secondary to UA. 

The present study has some limitations. First, baseline uric acid level measurements were used for analysis, and we did not obtain time-dependent changes. Second, our population was limited to the East Asian population in Taiwan, who have a high percentage of hyperuricemia [61]. We could not address hyperuricemia for renal outcome in other races. Third, as this was an observational study, causal relationships between MS components and hyperuricemia could not be confirmed. Fourth, dietary factors were not included in our study, which may also be crucial to renal outcome and prognosis. The last, most of the deceased were CVD patients, and fewer than ten were cancer patients. Some patients had left our two hospitals, and some of them had gone to the outside ones for dialysis. The deaths of patients who had left could only be traced but the cause of death could not be explored, and 30% of all-cause mortality is after dialysis and the cause of death is not clear, so we could not specifically examine the association between uric acid levels and cardiovascular mortality. We have arranged for more studies about the relationship between uric acid levels and cardiovascular mortality in the future.

## 5. Conclusions

Hyperuricemia is associated with worse renal outcome in non-MS/DM and is not an independent component of MS in CKD stage 1–4 patients. Both lines of evidence suggest that hyperuricemia secondary to MS could not predict renal outcome. We suggest that future clinical trials of ULA for renal outcome should be conducted in non-MS/DM populations.

## Figures and Tables

**Figure 1 biomedicines-10-01719-f001:**
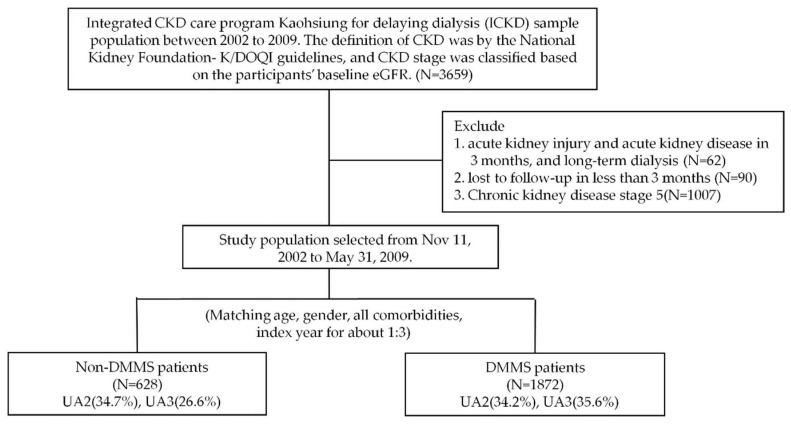
Flow diagram of the study population.

**Table 1 biomedicines-10-01719-t001:** Baseline characteristics of CKD stage 1–4 patients divided by MS/DM and uric acid tertiles.

	Non-MS/DM					MS/DM				
	All	UA			*p*	All	UA			*p*
Variable		Tertile 1	Tertile 2	Tertile 3			Tertile 1	Tertile 2	Tertile 3	
No. of patients	628	243 (38.7%)	218 (34.7%)	167 (26.6%)	*-*	1872	565 (30.2%)	640 (34.2%)	667 (35.6%)	*-*
UA range (mg/dL)										
Male		<7	7–8.5	≥8.5			<7	7–8.5	≥8.5	
Female		<6.4	6.4–8	≥8			<6.4	6.4–8	≥8	
Demographics										
Age (year)	58.0 (17.3)	56.3 (17.1)	59.5 (15.7)	58.5 (19.4)	<0.001	63.9 (13.1) ^#,$^	63.4 (12.8)	63.8 (12.8)	64.3 (13.8)	<0.001
Sex (female)	202 (32.2%)	94 (38.7%)	67 (30.7%)	41 (24.6%)	0.009	697 (37.2%) ^#,$^	205 (36.3%)	233 (36.4%)	259 (38.8%)	0.567
Comorbidity										
CVD	84 (13.4%)	31 (12.8%)	30 (13.8%)	23 (13.8%)	0.937	469 (25.1%) ^#,$^	122 (21.6%)	158 (24.7%)	189 (28.3%)	0.024
Hypertension	263 (41.9%)	87 (35.8%)	102 (46.8%)	74 (44.3%)	0.044	1250 (66.8%) ^#,$^	356 (63.0%)	441 (68.9%)	453 (67.9%)	0.070
Hyperuricemia	106 (16.9%)	21 (8.6%)	37 (17.0%)	48 (28.7%)	<0.001	342 (18.3%) ^$^	68 (12.0%)	114 (17.8%)	160 (24.0%)	<0.001
Severe liver disease	37 (5.9%)	13 (5.3%)	15 (6.9%)	9 (5.4%)	0.745	85 (4.5%)	20 (3.5%)	33 (5.2%)	32 (4.8%)	0.374
Cancer	53 (8.4%)	22 (9.1%)	21 (9.6%)	10 (6.0%)	0.403	137 (7.3%)	38 (6.7%)	49 (7.7%)	50 (7.5%)	0.806
Diuretics use	78 (12.4%)	25 (10.3%)	20 (9.2%)	33 (19.8%)	0.003	390 (20.8%) ^#^	91 (16.1%)	143 (22.3%)	156 (23.4%)	0.004
Renal function status										
eGFR (mL/min/1.73 m^2^)	46.1 (28.2)	56.1 (33.6)	42.7 (22.6)	36.1 (20.6)	<0.001	38.8 (21.1) ^#^	46.0 (25.7)	38.0 (19.2)	33.4 (16.1)	<0.001
UPCR (mg/g)	451 (162–1171)	423 (138–972)	432 (189–1131)	541 (188–1437)	0.318	793 (276–2063) ^#,$^	691 (233–1878)	837 (291–2408)	803 (307–1947)	0.031
Laboratory data										
Hemoglobin (g/dL)	12.4 (2.1)	12.5 (2.1)	12.5 (2.1)	12.2 (2.1)	<0.001	12.1 (2.2)	12.4 (2.3)	12.0 (2.1)	11.9 (2.2)	<0.001
Albumin (g/dL)	4.0 (0.5)	4.1 (0.5)	4.0 (0.5)	3.9 (0.5)	0.010	3.9 (0.5) ^#^	3.9 (0.6)	3.9 (0.5)	3.9 (0.5)	0.567
ALT (mg/dL)	23.6 (17.2)	24.2 (20.0)	23.5 (16.3)	22.9 (13.9)	0.308	27.8 (26.2)	27.9 (23.9)	26.4 (23.7)	29.2 (30.1)	0.954
CRP (mg/L)	0.7 (0.2–2.8)	0.6 (0.2–2.3)	0.6 (0.2–3.8)	0.8 (0.3–3.2)	0.389	1.1 (0.4–5.1) ^#,$^	1.0 (0.3–5.0)	1.2 (0.4–5.0)	1.3 (0.4–5.6)	0.099
Phosphorus (mg/dL)	3.7 (0.8)	3.6 (0.6)	3.7 (0.7)	4.0 (1.0)	<0.001	3.9 (0.8) ^#^	3.8 (0.8)	3.9 (0.8)	4.1 (0.9)	<0.001
Bicarbonate (mEq/L)	24.0 (3.8)	24.5 (3.8)	24.1 (3.6)	23.2 (3.8)	0.670	23.6 (3.7)	24.1 (3.9)	23.4 (3.7)	23.4 (3.5)	0.599
Uric acid (mg/dL)	7.2 (1.9)	5.5 (1.0)	7.4 (0.5)	9.5 (1.2)	<0.001	7.7 (2.0) ^#,$^	5.7 (0.9)	7.5 (0.5)	9.7 (1.5)	<0.001
Outcomes										
RRT	88 (14.0%)	17 (7.0%)	33 (15.1%)	38 (22.8%)	<0.001	459 (24.5%) ^#^	111 (19.6%)	172 (26.9%)	176 (26.4%)	0.009
RRT + mortality before RRT	163 (26.0%)	42 (17.3%)	60 (27.5%)	61 (36.5%)	<0.001	794 (42.4%) ^#,$^	197 (34.9%)	278 (43.4%)	319 (47.8%)	0.021
All-cause mortality	78 (12.4%)	25 (10.3%)	28 (12.8%)	26 (15.6%)	0.223	430 (23.0%) ^#,$^	120 (21.2%)	136 (21.3%)	174 (26.1%)	0.058

Abbreviations. UA, uric acid; CKD, chronic kidney disease; MS, metabolic syndrome; DM, diabetes mellitus; CVD, cardiovascular disease; eGFR, estimated glomerular filtration rate; UPCR, urine protein-to-creatinine ratio; ALT, alanine aminotransferase; CRP, C-reactive protein; RRT, renal replacement therapy. Data are presented as mean (standard error), median (interquartile range), or count (percentage%). #: *p* < 0.05 compared with all MS/DM group; $: *p* < 0.05 compared with UA tertile 3 in MS/DM group. Components of MS: 1. waist circumference ≥90 cm in men or ≥80 cm in women; 2. systolic blood pressure ≥130 mmHg or diastolic.

**Table 2 biomedicines-10-01719-t002:** MS components are divided by MS/DM and uric acid tertiles.

	Non-MS/DM					MS/DM				
	All	UA			*p*	All	UA			*p*
Variable		Tertile 1	Tertile 2	Tertile 3			Tertile 1	Tertile 2	Tertile 3	
Components of MS										
MS scores	1.5 (0.7)	1.4 (0.7)	1.5 (0.6)	1.5 (0.6)	0.001	3.6 (0.9)	3.4 (0.9)	3.6 (1.0)	3.7 (0.9)	0.710
Waist criteria1	133 (21.2%)	44 (18.1%)	48 (22.0%)	41 (24.6%)	0.272	1287 (68.8%)	352 (62.3%)	442 (69.1%)	493 (73.9%)	<0.001
BP criteria2	425 (67.7%)	151 (62.1%)	150 (68.8%)	124 (74.3%)	0.033	1690 (90.3%)	497 (88.0%)	585 (91.4%)	608 (91.2%)	0.084
HDL cholesterol criteria3	167 (26.6%)	62 (25.5%)	58 (26.6%)	47 (28.1%)	0.839	1232 (65.8%)	337 (59.6%)	437 (68.3%)	458 (68.7%)	0.001
Blood sugar criteria4	130 (20.7%)	58 (23.9%)	54 (24.8%)	18 (10.8%)	0.001	1559 (83.3%)	470 (83.2%)	546 (85.3%)	543 (81.4%)	0.167
TG criteria5	69 (11.0%)	21 (8.6%)	23 (10.6%)	25 (15.0%)	0.128	895 (47.8%)	259 (45.8%)	288 (45.0%)	348 (52.2%)	0.018
Associated data										
Waist (cm)	80.8 (11.1)	78.2 (11.2)	82.2 (10.9)	82.7 (10.5)	<0.001	91.1 (12.2)	89.2 (12.9)	91.7 (12.0)	92.2 (11.7)	<0.001
Systolic BP (mmHg)	131.1 (18.9)	128.8 (18.4)	130.7 (18.8)	134.9 (19.1)	0.005	140.1 (19.9)	139.4 (19.9)	141.0 (19.8)	139.8 (20.1)	0.575
Diastolic BP (mmHg)	79.3 (12.6)	77.8 (11.4)	79.9 (12.4)	80.6 (14.2)	<0.001	80.4 (12.8)	80.3 (12.2)	80.9 (12.4)	80.0 (13.6)	<0.001
Total cholesterol (mg/dL)	201.8 (55.7)	198.9 (56.0)	205.2 (54.5)	201.6 (56.7)	0.728	201.5 (58.8)	205.7 (68.1)	199.4 (49.9)	199.9 (58.0)	0.166
TG (mg/dL)	106.7 (59.0)	100.7 (47.5)	107.1 (64.8)	115.0 (65.0)	<0.001	177.9 (169.0)	171.6 (185.9)	173.8 (182.5)	187.2 (137.2)	<0.001
HDL cholesterol (mg/dL)	50.9 (15.2)	51.7 (15.4)	51.0 (14.7)	49.8 (15.5)	0.291	41.0 (12.4)	42.6 (13.5)	40.1 (11.8)	40.6 (12.0)	0.230
Blood glucose (mg/dL)	94.7 (16.0)	94.5 (15.1)	96.5 (20.0)	92.7 (10.3)	0.001	127.0 (50.7)	128.8 (52.5)	126.9 (50.1)	125.5 (49.8)	0.454
HbA1c (%)	5.5 (0.6)	5.5 (0.6)	5.6 (0.5)	5.5 (0.6)	0.377	7.0 (1.8)	7.2 (2.0)	7.0 (1.7)	6.9 (1.7)	0.343

Abbreviations. BP, blood pressure; TG, triglyceride; HDL, high-density lipoprotein; LDL, low-density lipoprotein, HbA1c: glycated hemoglobin; other abbreviations are the same as in Table 1. Data are presented as in Table 1.

**Table 3 biomedicines-10-01719-t003:** Multivariate linear regression for UA. (Full adjusted model).

Variables	β Coefficient	95% CI β Coefficient	*p*
constant	6.979		
Age (years)	−0.004	−0.009 to 0.002	0.172
Gender (female vs. male)	−0.400	−0.575 to −0.226	<0.001
eGFR (mL/min/1.73 m^2^)	−0.021	−0.025 to −0.018	<0.001
Upcr log	−0.008	−0.156 to 0.140	0.915
CVD	0.060	−0.120 to 0.240	0.512
HbA1c (%)	−0.200	−0.362 to −0.039	0.015
Smoker	0.043	−0.170 to 0.256	0.692
Cancer	−0.108	−0.372 to 0.155	0.421
Severe liver disease	0.219	−0.103 to 0.541	0.182
ULA	0.933	0.760 to 1.106	<0.001
Diuretics	0.364	0.153 to 0.574	0.001
Hypertension	−0.004	−0.155 to 0.147	0.955
Hemoglobin (g/dL)	0.019	−0.025 to 0.062	0.400
Cholesterol log	−0.777	−1.574 to 0.019	0.056
Albumin (g/dL)	−0.043	−0.204 to 0.117	0.594
CRP ln	0.125	0.046 to 0.205	0.002
Phosphorus (mg/dL)	0.296	0.202 to 0.391	<0.001
BMI (kg/m^2^)	0.034	0.010 to 0.058	0.006
MS components			
Waist (cm)	0.008	0.000 to 0.015	0.049
Mean BP (mmHg)	0.000	−0.006 to 0.005	0.947
HDL cholesterol (mg/dL)	0.004	−0.003 to 0.010	0.268
Blood glucose (mg/dL)	0.001	−0.001 to 0.003	0.217
TG log	0.420	0.058 to 0.781	0.023

Abbreviations. ULA, urate lowering agents; ln, natural log; BMI, body mass index; other abbreviations are the same as in Table 1 and Table 2. Data are presented as in Table 1.

**Table 4 biomedicines-10-01719-t004:** HR of UA for clinical outcomes divided by MS/DM and UA.

	Non-MS/DM				MS/DM			
	Uric Acid (per 1 mg/dL)	Uric Acid			Uric Acid (per 1 mg/dL)	Uric Acid		
Variables		Tertile 1	Tertile 2	Tertile 3		Tertile 1	Tertile 2	Tertile 3
HR for RRT			aHR (95% CI)	aHR (95% CI)			aHR (95% CI)	aHR (95% CI)
Unadjusted	1.26 (1.14–1.40) **	1 (reference)	2.30 (1.28–4.14) *	3.89 (2.20–6.90) **	1.10 (1.05–1.15) **	1 (reference)	1.41 (1.11–1.79) *	1.51 (1.19–1.91) **
Fully adjusted	1.27 (1.10–1.47) *	1 (reference)	2.30 (1.21–4.36) *	3.86 (1.87–7.97) **	0.96 (0.92–1.01)	1 (reference)	1.02 (0.79–1.31)	1.00 (0.77–1.30)
HR for RRT + 50% decline								
Unadjusted	1.18 (1.07–1.29) **	1 (reference)	1.45 (0.97–2.17)	1.85 (1.22–2.81) *	1.06 (1.02–1.10) *	1 (reference)	1.37 (1.13–1.67) *	1.28 (1.05–1.57) *
Fully adjusted	1.13 (1.04–1.24) *	1 (reference)	1.51 (1.00–2.29)	1.69 (1.08–2.65) *	0.98 (0.94–1.02)	1 (reference)	1.02 (0.83–1.25)	0.95 (0.77–1.17)
HR for RRT + Mortality before RRT								
Unadjusted	1.23 (1.13–1.33) **	1 (reference)	1.73 (1.16–2.58) *	2.53 (1.70–3.78) **	1.11 (1.07–1.15) **	1 (reference)	1.27 (1.06–1.52) *	1.53 (1.28–1.83) **
Fully adjusted	1.18 (1.06–1.31) *	1 (reference)	1.60 (1.05–2.45) *	1.73 (1.07–2.80) *	1.01 (0.98–1.04)	1 (reference)	1.06 (0.87–1.28)	1.20 (0.99–1.46)
HR for all-cause mortality								
Unadjusted	1.18 (1.05–1.32) *	1 (reference)	1.32 (0.77–2.28)	1.62 (0.93–2.83)	1.09 (1.04–1.14) **	1 (reference)	0.98 (0.77–1.25)	1.30 (1.03–1.64)
Fully adjusted	1.08 (0.93–1.25)	1 (reference)	1.39 (0.78–2.48)	0.98 (0.53–1.80)	1.05 (1.00–1.10) *	1 (reference)	0.92 (0.72–1.19)	1.18 (0.92–1.52)

Abbreviations. HR, hazard ratio; aHR, adjusted HR; CI, confidence interval; other abbreviations are the same as in Table 1, Table 2 and Table 3. Fully adjusted model: adjusted for age, sex, eGFR, log UPCR, CVD, HbA1c, smoker, cancer, severe liver disease, smoker, ULA, diuretics, HTN, Hb, albumin, ln CRP, phosphorus, BMI, waist, mean BP, HDL cholesterol, blood sugar, and log TG. Values expressed as aHR and 95% CI, *: *p* value < 0.05, **: *p* value < 0.01.

**Table 5 biomedicines-10-01719-t005:** HR of MS for clinical outcomes by tradition definition or adding UA in the definition.

	Traditional Definition		UA as 6th Criteria	
	Non-MS	MS	Non-MS	MS
HR for RRT		aHR (95% CI)		aHR (95% CI)
Unadjusted	1 (reference)	1.79 (1.47–2.18) **	1 (reference)	1.96 (1.56–2.45) **
Fully adjusted	1 (reference)	1.73 (1.24–2.43) *	1 (reference)	1.27 (0.94–1.71)
HR for RRT + 50% decline				
Unadjusted	1 (reference)	1.68 (1.43–1.97) **	1 (reference)	1.68 (1.41–2.01) **
Fully adjusted	1 (reference)	1.35 (1.12–1.63) *	1 (reference)	1.16 (0.95–1.41)
HR for RRT + Mortality before RRT				
Unadjusted	1 (reference)	1.66 (1.43–1.92) **	1 (reference)	1.74 (1.48–2.05) **
Fully adjusted	1 (reference)	1.28 (1.00–1.64) *	1 (reference)	1.17 (0.94–1.45)
HR for all-cause mortality				
Unadjusted	1 (reference)	1.55 (1.27–1.89) **	1 (reference)	1.51 (1.21–1.87) **
Fully adjusted	1 (reference)	1.00 (0.72–1.40)	1 (reference)	0.88 (0.65–1.19)

Abbreviations are the same as in Table 1, Table 2, Table 3 and Table 4. Fully adjusted model: adjusted for age, sex, eGFR, log UPCR, CVD, HbA1c, smoker, cancer, severe liver disease, smoker, ULA, diuretics, HTN, Hb, albumin, ln CRP, phosphorus, BMI, waist, mean BP, HDL cholesterol, blood sugar, and log TG. Values expressed as aHR and 95% CI, *: *p* value < 0.05, **: *p* value < 0.01.

## Data Availability

Not applicable.

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
