# Peer review of "Hyperuricemia, a Non-Independent Component of Metabolic Syndrome, Only Predicts Renal Outcome in Chronic Kidney Disease Patients without Metabolic Syndrome or Diabetes"

_biomedicines, 2022, doi:10.3390/biomedicines10071719_

Round 1

Reviewer 1 Report

The authors present a well-conducted study regarding the role of hyperuricemia in the progression of CKD in patients with and without MS. The results are well-presented and the manuscript well written. I have few comments regarding the manuscript:

1. The authors state that in the multivariate analysis among the adjusted covariates they used the term CVD (possibly cardiovascular disease?). How CVD was defined in this study? Please add under each table the covariates used for the multivariate analysis.

2. Since blood glucose and lipid profile changed among UA tertiles, why these covariates were not included in the multivariate analysis?

3. Please provide the criteria for MS used applied in this study.

4. In the Introduction section it is important to note that hyperuricemia is associated with obsesity in both non CKD and CKD patients. Please add these recent articles in the reference list;

Sun J, Yue C, Liu Z, Li J, Kang W. The Association Between Total Percent Fat and Serum Uric Acid in Adults. Front Nutr. 2022 May 20;9:851280.

Karava V, Dotis J, Kondou A, Christoforidis A, Liakopoulos V, Tsioni K, Kollios K, Papachristou F, Printza N. Association between relative fat mass, uric acid, and insulin resistance in children with chronic kidney disease. Pediatr Nephrol. 2021 Feb;36(2):425-434. 

Author Response

Q1: The authors state that in the multivariate analysis among the adjusted covariates they used the term CVD (possibly cardiovascular disease?). How CVD was defined in this study? Please add under each table the covariates used for the multivariate analysis. 

A1: Thanks for the reviewer’s opinion. CVD, cardiovascular disease, includes coronary artery disease and cerebrovascular disease (Line 86-87). We added it under table 4 and 5 as the covariates used for the multivariate analysis.

Q2: Since blood glucose and lipid profile changed among UA tertiles, why these covariates were not included in the multivariate analysis?

A2: Thanks for the reviewer’s suggestion. We added blood sugar, HbA1c, HDL cholesterol, and TG log as covariates of table 3, 4, and 5 for the multivariate analysis.

Q3: Please provide the criteria for MS used applied in this study. 

A3: I’m appreciated for the reviewer’s suggestion. Diagnosing metabolic syndrome (MS) according to the American Heart Association/National Heart, Lung and Blood Institute criteria. When a person has any of the following five abnormalities greater than or equal to 3: (Waist circumference greater than or equal to 102 cm for men, and greater than or equal to 88 cm for women, Asian American waist circumference greater than or equal to 90 cm for men and greater than or equal to 80 cm for women). Hypertriglyceridemia (triglycerides greater than or equal to 150 mg/dL or subject is taking lipid-controlling medications), low HDL (HDL; less than 40 mg/dL in men, less than50 mg/dL in women, or subject is taking lipid-controlling medications), hypertension (systolic blood pressure greater than or equal to 130 mmHg/diastolic blood pressure greater than or equal to 85 mmHg, or subject is taking hypertension medication), and hyperglycemia (fasting blood glucose greater than or equal to 100 mg/dL or subject is taking diabetes medication)[1]. As Taiwan is an Asian country, central obesity in Taiwanese national MS criteria was defined as waist circumference ≥90 cm in men and ≥80 cm in women by the Taiwan Department of Health[2]. Moreover, HDL is not included in the health check parameters mandated by occupational health regulations; therefore, high total cholesterol (≥200 mg/dL or taking lipid-controlling medication) was the criterion used instead of low HDL value. (Line 104-121)

Q4: In the Introduction section it is important to note that hyperuricemia is associated with obesity in both non-CKD and CKD patients. Please add these recent articles in the reference list;

Sun J, Yue C, Liu Z, Li J, Kang W. The Association Between Total Percent Fat and Serum Uric Acid in Adults. Front Nutr. 2022 May 20;9:851280.

Karava V, Dotis J, Kondou A, Christoforidis A, Liakopoulos V, Tsioni K, Kollios K, Papachristou F, Printza N. Association between relative fat mass, uric acid, and insulin resistance in children with chronic kidney disease. Pediatr Nephrol. 2021 Feb;36(2):425-434.

A4: I’m appreciated for the reviewer’s opinion. J. Sun, et al unveiled that a significant positive association between total percent fat and serum UA, especially in male population[3]. V. Karava, et al. found that even in children with high relative fat mass including normal weight obesity are at risk for IR regardless of chronic kidney disease (CKD) stage[4].  (Line 45-49)

Reference

  1. Grundy, S.M.; Cleeman, J.I.; Daniels, S.R.; Donato, K.A.; Eckel, R.H.; Franklin, B.A.; Gordon, D.J.; Krauss, R.M.; Savage, P.J.; Smith Jr, S.C. Diagnosis and management of the metabolic syndrome: an American Heart Association/National Heart, Lung, and Blood Institute scientific statement. Circulation 2005, 112, 2735-2752.
  2. Yeh, W.-C.; Chuang, H.-H.; Lu, M.-C.; Tzeng, I.-S.; Chen, J.-Y. Prevalence of metabolic syndrome among employees of a taiwanese hospital varies according to profession. Medicine 2018, 97.
  3. Sun, J.; Yue, C.; Liu, Z.; Li, J.; Kang, W. The Association Between Total Percent Fat and Serum Uric Acid in Adults. Frontiers in Nutrition 2022, 9.
  4. Karava, V.; Dotis, J.; Kondou, A.; Christoforidis, A.; Liakopoulos, V.; Tsioni, K.; Kollios, K.; Papachristou, F.; Printza, N. Association between relative fat mass, uric acid, and insulin resistance in children with chronic kidney disease. Pediatric Nephrology 2021, 36, 425-434.

Reviewer 2 Report

In this manuscript, authors reported that higher serum uric acid level was associated with poor renal outcome defined by the initiation of renal replacement therapy in chronic kidney disease patients without metabolic syndrome and diabetes mellitus in a single center prospective study. The subject of study seems to be interesting. However, there are some major concerns in this study. The reviewer’s comments are described as follows.

1. In this study, authors analyzed all-cause mortality. However, hyperuricemia is generally known to be related to cardiovascular mortality and significant subjects with cancers were included in this study. Thus, authors should specifically examine the association between uric acid levels and cardiovascular mortality rather than all-cause mortality.

2. Authors defined study outcomes as renal replacement therapy and /or all-cause mortality. However, this study included patients with chronic kidney disease stage 1 to 4. Generally considering, it is extremely improbable that patients with chronic kidney disease stage 1 could initiate renal replacement therapy during the study period, and thus such study outcome may be appropriate for patients with chronic kidney disease stage 3 and 4 but not for stage 1 and 2. If authors include all subjects in this study, loss of renal function (determined by doubling serum creatinine or 50% decrease in estimated GFR, for example) should be added as an alternative study outcome.

3. Authors have to consider the effects of antihyperuricemic drugs on renal function and outcomes. Especially, patients with metabolic syndrome and diabetes are tended to be treated with antihyperuricemic drugs. In addition, it is difficult to interpret the results because patients with hyperuricemia could be treated with any medications during the study period. Management of hyperuricemia throughout the study period rather that baseline uric acid levels might be important to affect study outcomes.

4. In the method section, the definitions of metabolic syndrome and diabetes mellitus in this study must be described.

5. In the study design, why and how authors selected 2500 subjects from 3659 patients. Authors have to make the flow chart for explaining the reasons.

Author Response

Q1: In this study, authors analyzed all-cause mortality. However, hyperuricemia is generally known to be related to cardiovascular mortality and significant subjects with cancers were included in this study. Thus, authors should specifically examine the association between uric acid levels and cardiovascular mortality rather than all-cause mortality. 

A1: Thanks for the reviewer’s opinion. Most of the deceased were CVD patients, and fewer than ten were cancer patients. Some patients had left our two hospitals, and some of them had gone to the outside ones for dialysis. The deaths of patients who had left could only be traced but the cause of death couldn’t be explored, so we could not specifically examine the association between uric acid levels and cardiovascular mortality.

Q2: Authors defined study outcomes as renal replacement therapy and /or all-cause mortality. However, this study included patients with chronic kidney disease stage 1 to 4. Generally considering, it is extremely improbable that patients with chronic kidney disease stage 1 could initiate renal replacement therapy during the study period, and thus such study outcome may be appropriate for patients with chronic kidney disease stage 3 and 4 but not for stage 1 and 2. If authors include all subjects in this study, loss of renal function (determined by doubling serum creatinine or 50% decrease in estimated GFR, for example) should be added as an alternative study outcome.

A2: Thanks for the reviewer’s suggestion. We added 50% decrease in estimated GFR as outcome analysis in table 4 and table 5. In the non-MS/DM group, in the fully adjusted Cox regression (Table 4), a 1 mg/dl increase in UA was significantly related to a 13% increase in the risk of RRT + 50% eGFR decline (HR: 1.13, 95% CI: 1.04-1.24) (Line 179-180). UA tertile 3 had a higher risk of RRT + 50% eGFR decline (HR: 1.69, 95% CI: 1.08-2.65) compared with UA tertile 1 (Line 182-183). In the MS/DM group, in the fully adjusted Cox regression (Table 4), the level of UA was not related to RRT + 50% eGFR decline (Line 187). To study whether UA could be a component of MS, we compared the association between clinical outcomes and MS (defined by traditional 3 of 5 components or by UA as 6th components [3 of 6 components]) (Table 5). In the fully adjusted Cox regression, MS by traditional definition was related to a 35% increase in the risk of RRT + 50% eGFR decline (HR: 1.35, 95% CI: 1.12-1.63) (Line 221-222). By adding UA in the definition, MS did not relate to RRT + 50% eGFR decline (Line 224).

Q3: Authors have to consider the effects of antihyperuricemic drugs on renal function and outcomes. Especially, patients with metabolic syndrome and diabetes are tended to be treated with antihyperuricemic drugs. In addition, it is difficult to interpret the results because patients with hyperuricemia could be treated with any medications during the study period. Management of hyperuricemia throughout the study period rather that baseline uric acid levels might be important to affect study outcomes. 

A3: I’m appreciated for the reviewer’s suggestion. We added urate lowering agents (ULA) as multivariate analysis in table 3 (Line 205) and outcome analysis in table 4 (Line 214) and 5 (Line 226). The use of ULA and diuretics was significantly related to higher uric acid levels (Line 171).

Q4: In the method section, the definitions of metabolic syndrome and diabetes mellitus in this study must be described.

A4: I’m appreciated for the reviewer’s opinion. (Line 103-126)

Diagnosing metabolic syndrome (MS) according to the American Heart Association/National Heart, Lung and Blood Institute criteria. When a person has any of the following five abnormalities greater than or equal to 3: (Waist circumference greater than or equal to 102 cm for men, and greater than or equal to 88 cm for women, Asian American waist circumference greater than or equal to 90 cm for men and greater than or equal to 80 cm for women). Hypertriglyceridemia (triglycerides greater than or equal to 150 mg/dL or subject is taking lipid-controlling medications), low HDL (HDL; less than 40 mg/dL in men, less than50 mg/dL in women, or subject is taking lipid-controlling medications), hypertension (systolic blood pressure greater than or equal to 130 mmHg/diastolic blood pressure greater than or equal to 85 mmHg, or subject is taking hypertension medication), and hyperglycemia (fasting blood glucose greater than or equal to 100 mg/dL or subject is taking diabetes medication)[1]. As Taiwan is an Asian country, central obesity in Taiwanese national MS criteria was defined as waist circumference ≥90 cm in men and ≥80 cm in women by the Taiwan Department of Health[2]. Moreover, HDL is not included in the health check parameters mandated by occupational health regulations; therefore, high total cholesterol (≥200 mg/dL or taking lipid-controlling medication) was the criterion used instead of low HDL value.

The diagnosis of diabetes was previously defined according to the 1999 World Health Organization (WHO) diagnostic criteria, fasting blood glucose ≥ 126 mg/dl, and two-hour blood glucose (postprandial) ≥ 200 mg/dl[3]. According to the American Diabetes Association (ADA) guidelines, an HbA1C level of 6.5% or higher is also in the range of diabetes[4].

Q5: In the study design, why and how authors selected 2500 subjects from 3659 patients. Authors have to make the flow chart for explaining the reasons.

A5: Thanks for the reviewer’s comments. Patients with CKD stages 1-5 who did not receive renal replacement therapy were included as inclusion criteria(N=3,659) at first, and 1. acute kidney injury, defined as >50% decrease in estimated glomerular filtration rate (eGFR) within 3 months, 2. lost to follow-up in less than 3 months (N=90), 3. chronic kidney disease stage 5(N=1,159), was excluded, then finally this analysis included 2500 patients with CKD stages 1-4 (Line 74-79). We made the flow chart of the study population for explaining the reasons as Figure 1 (as attachment).

Reference

  1. Grundy, S.M.; Cleeman, J.I.; Daniels, S.R.; Donato, K.A.; Eckel, R.H.; Franklin, B.A.; Gordon, D.J.; Krauss, R.M.; Savage, P.J.; Smith Jr, S.C. Diagnosis and management of the metabolic syndrome: an American Heart Association/National Heart, Lung, and Blood Institute scientific statement. Circulation 2005, 112, 2735-2752.
  2. Yeh, W.-C.; Chuang, H.-H.; Lu, M.-C.; Tzeng, I.-S.; Chen, J.-Y. Prevalence of metabolic syndrome among employees of a taiwanese hospital varies according to profession. Medicine 2018, 97.
  3. Alberti, K.G.M.M.; Zimmet, P.Z. Definition, diagnosis and classification of diabetes mellitus and its complications. Part 1: diagnosis and classification of diabetes mellitus. Provisional report of a WHO consultation. Diabetic medicine 1998, 15, 539-553.
  4. Færch, K.; Alssema, M.; Mela, D.J.; Borg, R.; Vistisen, D. Relative contributions of preprandial and postprandial glucose exposures, glycemic variability, and non-glycemic factors to HbA 1c in individuals with and without diabetes. Nutrition & diabetes 2018, 8, 1-9.

Round 2

Reviewer 2 Report

Authors have successfully addressed some of the reviewer's concerns. However, the following points have not been addressed in the revised manuscript.

Comment #1: The fact that authors could not specifically examine the association between uric acid and cardiovascular mortality appears to be a serious concern. This limitation has to be explained in the text in detail.

Comment #4: In Figure 1, number of subjects should be included in all columns.

Author Response

Authors have successfully addressed some of the reviewer's concerns. However, the following points have not been addressed in the revised manuscript.

Comment #1: The fact that authors could not specifically examine the association between uric acid and cardiovascular mortality appears to be a serious concern. This limitation has to be explained in the text in detail. 

A: Most of the deceased were cardiovascular disease patients, and fewer than ten were cancer patients. Some patients had left our two hospitals, and some of them had gone to the outside ones for dialysis. The deaths of patients who had left could only be traced but the cause of death couldn’t be explored, and 30% of all-cause mortality is after dialysis and the cause of death is not clear, so we could not specifically examine the association between uric acid levels and cardiovascular mortality. We have arranged for more studies about the relationship between uric acid levels and cardiovascular mortality in the future. (Line 322-329)

Comment #4: In Figure 1, number of subjects should be included in all columns.

A: Thanks for the reviewer’s comments. We have added the number of subjects in title (upper) column (N=3,659) and the column of exclude (N=62, 90, 1,007) in Figure 1. (Line 101-102)

Round 3

Reviewer 2 Report

There are no more comments.